# Hyperpolarised $^1$H–$^{13}$C Benchtop NMR Spectroscopy

**Alastair D. Robinson**, **Peter M. Richardson** and **Meghan E. Halse** *

Centre for Hyperpolarisation in Magnetic Resonance, Department of Chemistry, University of York, Heslington, York YO10 5DD, UK; adr518@york.ac.uk (A.D.R.); peter.richardson@york.ac.uk (P.M.R.)
* Correspondence: meghan.halse@york.ac.uk

**Abstract:** Benchtop NMR spectrometers with sub-ppm spectral resolution have opened up new opportunities for performing NMR outside of the standard laboratory environment. However, the relatively weak magnetic fields of these devices (1–2 T) results in low sensitivity and significant peak overlap in $^1$H NMR spectra. Here, we use hyperpolarised $^{13}$C{$^1$H} NMR to overcome these challenges. Specifically, we demonstrate the use of the signal amplification by reversible exchange (SABRE) *para*hydrogen-based hyperpolarisation technique to enhance the sensitivity of natural abundance 1D and 2D $^{13}$C{$^1$H} benchtop NMR spectra. We compare two detection methods for SABRE-enhanced $^{13}$C NMR and observe an optimal $^{13}$C{$^1$H} signal-to-noise ratio (SNR) for a refocused INEPT approach, where hyperpolarisation is transferred from $^1$H to $^{13}$C. In addition, we exemplify SABRE-enhanced 2D $^{13}$C benchtop NMR through the acquisition of a 2D HETCOR spectrum of 260 mM of 4-methylpyridine at natural isotopic abundance in a total experiment time of 69 min. In theory, signal averaging for over 300 days would be required to achieve a comparable SNR for a thermally polarised benchtop NMR spectrum acquired of a sample of the same concentration at natural abundance.

**Keywords:** NMR spectroscopy; benchtop; low-field; *para*hydrogen; hyperpolarisation; SABRE

## 1. Introduction

Benchtop NMR spectrometers have the potential to open up new applications for NMR spectroscopy outside of the traditional laboratory environment owing to the relative portability of these devices. Of particular interest are permanent magnet spectrometers with magnetic field strengths of 1–2 T that have the capability to record high-resolution NMR spectra [1]. Whilst several of the original NMR discoveries were made using permanent magnet NMR spectrometers [2–9], their usage became limited once strong and stable superconducting electromagnets became readily available [10–12]. In the early 2000s, the prospect of cheap, cryogen-free, and compact NMR spectrometers prompted the resurgence of permanent magnet-based systems [13]. A major obstacle to these spectrometers was the need for highly homogenous magnetic fields as high-resolution spectra require field spatial variations of less than tens-of-parts per billion [14]. It was through specifically designed Halbach arrays of magnets [15], advancements in shimming electronics, and temperature stabilisation that these tens-of-ppb magnetic field homogeneities were achieved [16–18]. From this point, a range of high-resolution benchtop NMR spectrometers with different field strengths and heteronuclear detection capabilities have been developed and implemented across a plethora of applications [19], such as industrial quality control [20–26], $^1$H and $^{13}$C NMR undergraduate teaching [27–32], and on-line reaction monitoring [33–39].

A significant limitation of these benchtop spectrometers is their low inherent sensitivity. All NMR experiments are considered to be insensitive due to the small Boltzmann population difference between the nuclear energy levels that form when NMR-active nuclei are placed within an external magnetic field. The energy level spacing is magnetic field strength dependent and so moving to lower magnetic

field strengths further reduces sensitivity [40]. Indeed, overall NMR sensitivity scales with magnetic field strength approximately proportional to $B_0^{3/2}$. Additionally, lower magnetic field strengths reduce spectral chemical shift dispersion, which scales linearly with field, and can cause strong coupling issues, where coupling constants are similar in magnitude to chemical shift differences between coupled spins. As a result, broad and overlapping peaks are common in low-field NMR spectra, even for simple molecules. This is a particular challenge of $^1$H benchtop NMR spectroscopy because of the relatively small chemical shift range of $^1$H nuclei.

To improve the general applicability of benchtop NMR spectrometers, novel methods are required to overcome the challenges of sensitivity and resolution. In this work, we explore the potential for natural abundance $^{13}$C NMR spectra to be used to surmount the issue of reduced resolution. Due to isotopic dilution, the increased chemical shift range, when compared to $^1$H NMR and the ability to simplify spectra through broadband $^1$H decoupling and natural abundance $^{13}$C NMR spectra, can be as readily interpreted at 43 MHz (1 T) as at 300 MHz (7 T). However, benchtop $^{13}$C NMR spectroscopy poses a significant sensitivity challenge because the receptivity of natural abundance $^{13}$C is $1.7 \times 10^{-4}$ relative to $^1$H.

In principle, the low sensitivity of benchtop NMR can be overcome using hyperpolarisation. Hyperpolarisation methods generate a population difference that is orders of magnitude larger than at thermal equilibrium and so provide large NMR signal enhancements. Popular hyperpolarisation techniques include Dynamic Nuclear Polarisation (DNP) [41–43] and *Para*hydrogen-induced polarisation (PHIP) [44–46]. One relatively inexpensive method, which has been successfully implemented at low-field, is a PHIP-based method called signal amplification by reversible exchange (SABRE) [47–49]. SABRE catalytically transfers the latent polarisation in *para*hydrogen ($p$-$H_2$, the NMR-silent singlet spin isomer of $H_2$) to a target molecule without chemical alteration of the target. The mechanism by which this occurs, a simplified scheme of which is shown in Figure 1, is through reversible binding of $p$-$H_2$ and the target analyte to a metal complex (commonly iridium-based) in the presence of a weak polarisation transfer field (PTF) in the range of 0–20 mT. Oxidative addition to the metal complex breaks the symmetry of the $p$-$H_2$ molecule, allowing for its stored polarisation to be transferred through the scalar coupling network of the complex to the target molecule that is also bound to the metal. Both the $p$-$H_2$ and the target analyte bind reversibly, leading to a buildup of hyperpolarised analyte in solution over a period of seconds [50].

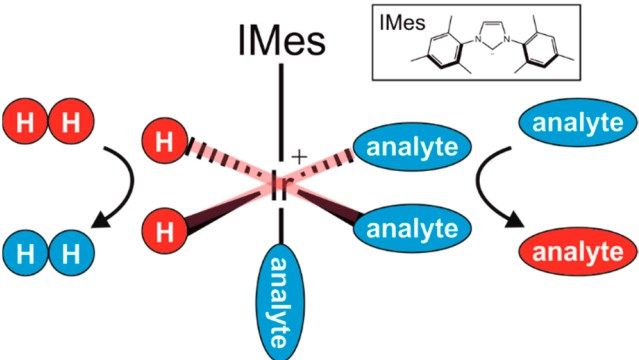

**Figure 1.** A schematic representation of the signal amplification by reversible exchange (SABRE)-polarisation transfer process. Hyperpolarisation is catalytically transferred from the nuclear singlet isomer of hydrogen, *para*hydrogen ($p$-$H_2$), to the target analyte via the *J* coupling network of the active SABRE catalyst. Rapid reversible exchange of the analyte and $p$-$H_2$ on the catalyst leads to a buildup of hyperpolarised analyte in solution over a period of seconds. In general, this process is optimised in a weak magnetic field of 0–20 mT. The active SABRE catalyst is a positively charged Ir(III) di-hydride complex with two molecules of the analyte bound *trans* to the hydrides and a N-heterocyclic carbene (IMes, shown in the inset) bound *trans* to a third molecule of the analyte.

Many factors influence the effectiveness of the SABRE-polarisation transfer but of significant importance is the PTF [51]. The PTF determines the pathway of the magnetisation transfer through the coupling network [52]. The resonance condition that leads to optimal transfer varies significantly for $^1$H SABRE and for transfer to other nuclei [53]. In the SABRE-SHEATH (SABRE in Shield Enables Alignment Transfer to Heteronuclei) approach, the focus is polarisation transfer to heteronuclei, which requires a PTF below the Earth's magnetic field. In this case, a μ-metal shield coupled with an electromagnet is used to reach the microtesla PTF required for direct polarisation transfer to the heteronucleus (e.g., $^{15}$N) [54–56]. In principle, SABRE-SHEATH can be used for direct polarisation transfer to $^{13}$C; however, due to the coupling network within the active SABRE-polarisation transfer complex, this direct transfer is often highly inefficient [57]. Optimal $^{13}$C hyperpolarisation is typically achieved through an indirect transfer mediated by $^1$H or $^{15}$N [58]. In this work, we investigate the feasibility of SABRE-hyperpolarised benchtop $^{13}$C{$^1$H} NMR experiments with a focus on the optimisation of the pulse sequences used for hyperpolarised $^{13}$C detection and an analysis of the challenges and opportunities provided by hyperpolarised 2D $^{13}$C–$^1$H benchtop NMR spectroscopy.

## 2. Materials and Methods

SABRE-hyperpolarised NMR spectra are acquired by dissolving $H_2$ gas, enriched in the *para* state, in a solution containing the active SABRE catalyst and the target analyte within a weak polarisation transfer field (PTF) typically in the range of 0–20 mT [59,60]. The introduction of $H_2$ into the solution is achieved either manually, via sample shaking, or via bubbling. In the so-called shake-and-drop approach, the headspace of an NMR tube fitted with a Young's tap is filled with several bars of *p*-$H_2$. The sample is shaken vigorously within the desired polarisation transfer field (PTF) for a period of a few seconds, allowing for the buildup of hyperpolarised analyte in solution. This polarisation transfer step is followed by rapid manual transfer of the sample into the NMR spectrometer for signal detection. In the automated flow-based approach, *p*-$H_2$ is bubbled through the solution for a period of seconds in a reaction chamber that sits within a small electromagnet that provides the desired PTF [61,62]. Following polarisation transfer, the sample is flowed pneumatically under a pressure of $N_2$ gas into the NMR spectrometer for detection. In general, the automated approach provides a lower SABRE enhancement due to a combination of engineering limitations including less efficient mixing, longer sample transfer times, and lower levels of *p*-$H_2$ enrichment in solution during bubbling. However, the flow system provides control over parameters such as the duration of polarisation, PTF, and sample transfer time and therefore can generate the level of reproducibility required to achieve SABRE-enhanced 2D NMR spectroscopy. The level of reproducibility of this system has been assessed previously which gave experimental variation of ~5% [49]. In this work, the 1D NMR spectra acquired to investigate the efficacy of different $^{13}$C NMR detection strategies were carried out using the manual shaking approach and the SABRE-enhanced 2D spectroscopy was achieved using an automated flow system.

For manual SABRE experiments, the sample was made up in a NMR tube fitted with a Young's valve. The sample contained 260 mM of 4-methylpyridine (4-MP) and 5.2 mM of [IrCl(COD)(IMes)] pre-catalyst (where COD is 1,5-cyclooctadiene and IMes is 1,3-bis(2,4,6-trimethyl-phenyl)-imidazolium) made up to 0.6 mL with methanol-$d_4$. The sample was de-gassed under vacuum using a freeze–pump–thaw method (detailed by Shaver et al. [63] but with liquid $N_2$ replaced with a dry-ice acetone bath) to allow the sample to be placed under an atmosphere of *p*-$H_2$ during SABRE experiments. Repeat shake-and-drop experiments were performed on a single sample by evacuating the head space and refilling with *p*-$H_2$ between experiments. *Para*hydrogen was generated by cooling $H_2$ gas over a paramagnetic catalyst based on activated charcoal at 28 K (with a conversion efficiency of 99%). The design of this generator has been described previously in Reference [64]. A handheld magnetic array with a 6.1 mT field strength was used to supply the necessary PTF during SABRE transfer [65]. The sample shaking time was 10 s and the sample transfer time was 2.0 ± 0.5 s in all experiments. On the addition of *p*-$H_2$ to the solution, the pre-catalyst will convert to the active form,

[Ir(IMes)(H)$_2$(4-MP)$_3$]Cl. Full conversion to the active form is required prior to achieving quantitative SABRE results. The activation was monitored by acquiring 6 repeat $^1$H shake-and-drop experiments over a typical period of 10 min, with the addition of fresh *p*-H$_2$ to the headspace of the NMR tube between each experiment.

Automated SABRE experiments were performed on a 3 mL solution containing 4-methylpyridine (260 mM) and 5.2 mM of the pre-catalyst [IrCl(COD)(IMes)] in methanol-*d$_4$*. Samples were loaded into a flow system consisting of a mixing chamber held within an electromagnet capable of generating magnetic field strengths between 0 and 14 mT and a custom-designed flow cell that holds the sample within the benchtop NMR spectrometer in the detection region (see Figure S1 in the supporting information) [49]. These were connected by fluorinated ethylene propylene tubing with sample transference being controlled by a pneumatic control unit (Bruker) with a supply of N$_2$ gas (6 bar absolute). The *para*hydrogen used in the automated SABRE experiments was generated using the same home-built generator described above. More details on this automated SABRE system can be found in Reference [49]. SABRE hyperpolarisation was achieved by bubbling *p*-H$_2$ at 4 bar (absolute) through the sample within the mixing cell for a fixed period of time (15 s). The pressure was then released and N$_2$ gas was used to transfer the sample into the flow cell within the benchtop NMR spectrometer for detection. The sample transfer time, including a 3 s delay for the H$_2$ pressure release step, was 4.1 s. For multiple-step experiments, an additional inter-scan delay of 16 s was included to return the sample to the mixing chamber and to allow for relaxation and full recovery of the *p*-H$_2$ pressure. In a similar fashion to the manual shaking method, full conversion of the pre-catalyst to the active form must be completed before performing quantitative experiments. Eight $^1$H pulse-and-acquire experiments on the flow system were conducted over 15 min to monitor the activation process.

All NMR data were collected using a 43 MHz (1 T) NMR spectrometer (Spinsolve Carbon, Magritek, Aachen, Germany) equipped with $^1$H/$^{19}$F and $^{13}$C channels. At the start of each session, shimming and frequency calibrations were performed on a reference sample containing a 10%:90% H$_2$O:D$_2$O mixture. All non-hyperpolarised benchtop NMR spectra were performed using a 0.6 mL sample of neat 4-methylpyridine (10.3 M) in a standard 5 mm NMR tube. The NMR detection sequences employed in this work are illustrated in Figure 2. In principle, 180° refocusing pulses should be used in the INEPT and refocused PA pulse sequences to improve performance by refocusing the chemical shift evolution. However, it was found for both hyperpolarised and thermally polarised experiments on the benchtop NMR spectrometer, the presence of additional 180° pulses led to lower SNR. We attribute this effect to poor RF pulse homogeneity. For all hyperpolarisation experiments, these detection sequences were applied immediately following SABRE hyperpolarisation using one of the two methods detailed above. A list of the variable flip angles used in the single-shot hyperpolarisation lifetime experiments (Figure 2e) are provided in Table S1 in the supporting information. These flip angles were chosen such that each pulse excited a fixed fraction of the available magnetisation, enabling a fit of the resultant signal integrals to a simple exponential decay function in order to determine the hyperpolarisation lifetime. The reported values of $^1$H and $^{13}$C hyperpolarisation lifetimes are the average of five repeated measurements and the standard error across the repetitions was used to define the error bars. The PTFs used for the $^1$H and $^{13}$C hyperpolarisation lifetime measurements were 6.1 mT (as described above) and ~50 μT (the Earth's magnetic field), respectively. All spectra were processed and $^{13}$C SNR values were calculated using MestReNova (Mestrelab research).

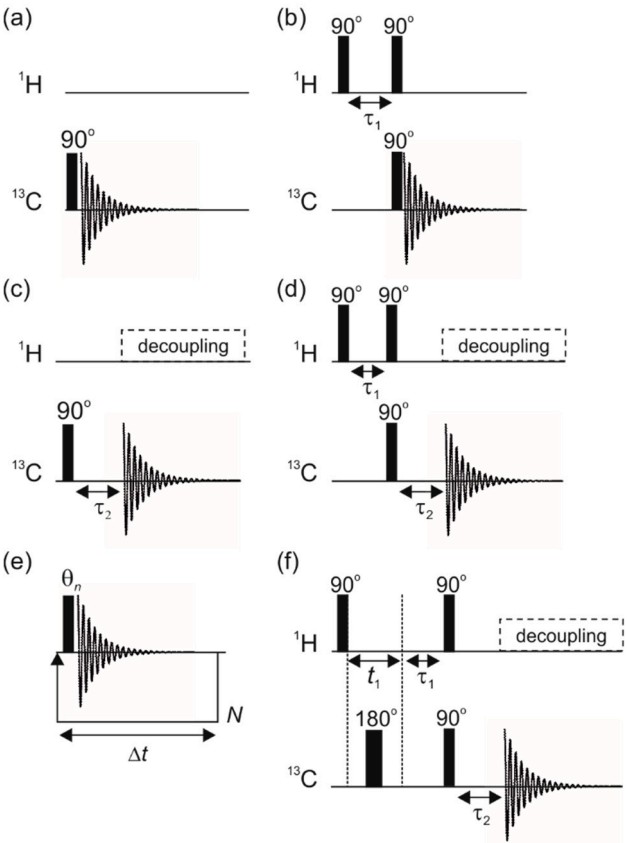

**Figure 2.** NMR pulse sequences where $\tau_1 = 1/2J_{CH}$ and $\tau_2 = 1/3J_{CH}$. (**a**) Pulse and acquire (PA) for direct $^{13}$C detection, (**b**) INEPT for transfer from $^1$H to $^{13}$C, (**c**) PA with refocusing delay and $^1$H decoupling, (**d**) INEPT with refocusing delay and $^1$H decoupling, (**e**) single-shot variable flip angle hyperpolarisation lifetime measurement (see Table S1 for values of $\theta_n$), and (**f**) 2D HETCOR, where $t_1$ is incremented to encode $^1$H chemical shift into the indirect dimension. For hyperpolarisation experiments, the pulse sequences are applied immediately following the SABRE-hyperpolarisation step and sample transfer into the benchtop NMR spectrometer.

## 3. Results

### 3.1. Optimal Detection of SABRE-Enhanced $^{13}$C Benchtop NMR Spectra

To explore the optimal detection approach for SABRE-hyperpolarised benchtop $^{13}$C NMR spectroscopy, we compared two methods for hyperpolarised signal acquisition. In the first pulse-and-acquire (PA) approach, the $^{13}$C NMR signal was detected directly following a single 90° excitation pulse (Figure 2a). SABRE-enhanced benchtop $^{13}$C NMR spectra acquired using PA have been reported previously [49]. Here we compared this approach with a second method where the $^{13}$C NMR signal was detected indirectly following a *J*-based INEPT transfer of hyperpolarisation from $^1$H to $^{13}$C (Figure 2b). A comparison of the SABRE-hyperpolarised $^{13}$C benchtop NMR spectra of 4-methylpyridine (4-MP) using these two approaches is presented in Figure 3a. In both cases, SABRE hyperpolarisation was achieved using the manual sample shaking method.

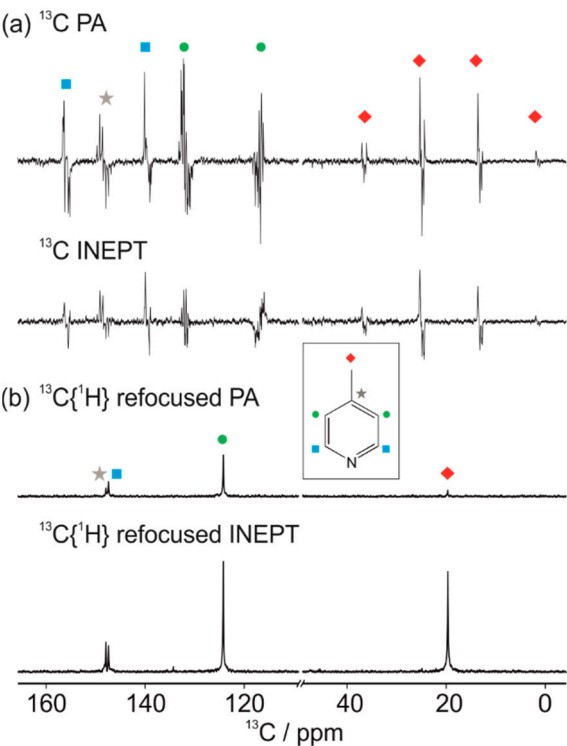

**Figure 3.** 1D SABRE-hyperpolarised $^{13}$C NMR spectra of 260 mM 4-methylpyridine at natural abundance in MeOD with 5.2 mM active SABRE catalyst. Each spectrum was acquired in a single scan on a benchtop (43 MHz) NMR spectrometer following SABRE hyperpolarisation in a PTF of 6.1 mT. (**a**) Fully-coupled $^{13}$C NMR spectra acquired immediately following a 90° $^{13}$C pulse (top) and following INEPT transfer from $^1$H with $J_{CH}$ = 10 Hz. (**b**) $^{13}$C{$^1$H} NMR spectra acquired using PA (top) and INEPT transfer from $^1$H (bottom). A refocusing delay of $(3J_{CH})^{-1}$ with $J_{CH}$ = 10 Hz was included prior to signal acquisition in both cases.

Inspection of the fully-coupled, SABRE-enhanced $^{13}$C NMR spectra in Figure 3a reveals that a higher overall $^{13}$C NMR signal enhancement was observed for the PA detection scheme (Figure 3a, top) when compared to the INEPT approach where polarisation was transferred from $^1$H to $^{13}$C (Figure 3b bottom). This suggests that the efficiency of the spontaneous indirect transfer from $^1$H to $^{13}$C during SABRE in the PTF is higher than the efficiency of the RF-driven transfer achieved by the INEPT sequence following transfer of the sample to the NMR detector. One potential explanation for the higher $^{13}$C signal observed in the PA experiment is relaxation. If the lifetime of the $^1$H polarisation is shorter than for the $^{13}$C hyperpolarisation, a higher proportion of available signal will decay during sample transport in the INEPT case when compared to the PA case. Figure 4 presents a comparison of the $^{13}$C and $^1$H hyperpolarisation lifetimes as a function of concentration of 4-methylpyridine, where the concentration of the catalyst is 5.2 mM in all cases. These single-shot lifetime measurements were acquired on the benchtop NMR spectrometer using the variable flip angle sequence in Figure 2e immediately following hyperpolarisation using the manual shaking SABRE procedure. Contrary to the hypothesis, we found that the lifetimes for the $^1$H hyperpolarisation are longer than for the directly detected $^{13}$C polarisation. Therefore, these results do not support the proposition that relaxation effects lead to higher observed $^{13}$C signals in the PA case. However, it should be noted that these experiments do not distinguish between $^1$H hyperpolarisation in molecules with and without $^{13}$C.

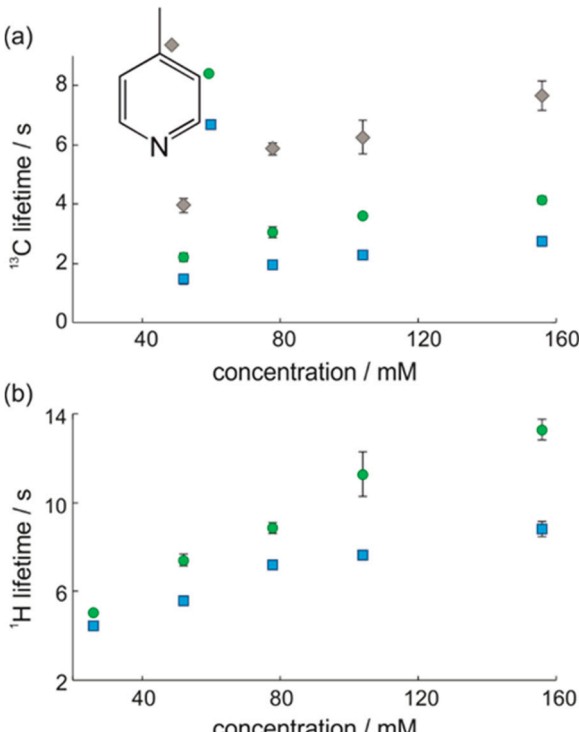

**Figure 4.** (**a**) $^{13}$C and (**b**) $^{1}$H SABRE-hyperpolarisation lifetimes measured using a single-shot pulse sequence (Figure 2e) at 1 T (43 MHz) for 4-methylpyridine in methanol-$d_4$ with 5.2 mM SABRE catalyst. Values are the average over five experiments and error bars are the standard error across the repeat measurements.

A second possible explanation for the lower signals observed following the INEPT transfer is that the efficiency of hyperpolarisation transfer to $^{1}$H is not optimised under these experimental conditions for molecules containing $^{13}$C. Both of the fully-coupled, SABRE-enhanced $^{13}$C NMR spectra in Figure 3a contained peaks that had anti-phase character with respect to a relatively long range $^{1}$H-$^{13}$C coupling on the order of $J_{CH}$ = 10 Hz. This anti-phase character is consistent with previous observations in the literature involving indirect SABRE-polarisation transfer to $^{13}$C via $^{1}$H [66]. In the case of the PA experiment, the anti-phase character indicated that the SABRE process had enhanced two-spin-order terms involving non-directly-bonded $^{1}$H–$^{13}$C pairs within the analyte [49]. In the case of the INEPT transfer experiment, we found that the optimal $^{13}$C NMR signal was observed using a constant of $J_{CH}$ = 10 Hz for the transfer step. Interestingly, the $^{13}$C NMR signal observed for a larger one-bond coupling constant of $J_{CH}$ = 140 Hz was reduced. This is in contrast to INEPT experiments carried out on samples at thermal equilibrium, where the most efficient transfer of polarisation is achieved between directly bonded $^{1}$H and $^{13}$C nuclei. These results indicate that protons directly bonded to $^{13}$C are less efficiently hyperpolarised via the SABRE process under our experimental conditions.

The lower efficiency of $^{1}$H hyperpolarisation for protons directly bonded to $^{13}$C can be understood by considering the resonance condition that facilitates spontaneous polarisation transfer in SABRE. Efficient polarisation transfer requires that the difference in chemical shift between the source of the polarisation (the $p$-H$_2$-derived hydrides in the polarisation transfer complex) and the target nuclei of the analyte bound to the complex be approximately equal to the dominant $J$ coupling constant within the coupling network. Typically, this is $J_{HH} \sim 8$ Hz between the pair of hydrides. This gives rise to an optimal PTF for transfer to aromatic $^{1}$H resonances in the analyte of ~6.5 mT and of a few µT for direct transfer to other nuclei such as $^{15}$N and $^{13}$C. However, if the protons in the target analyte are directly bonded to $^{13}$C, the dominant coupling will be the one-bond $^{13}$C–$^{1}$H coupling on the order of 140 Hz. This will significantly shift the resonance condition to a PTF on the order of ~0.1 T. Therefore, polarisation transfer to protons directly bonded to $^{13}$C is inefficient in the PTF of 6.1 mT used in these

experiments. This effect has been observed previously in the case of SABRE hyperpolarisation of $^{13}$C in acetonitrile, where no polarisation transfer to the methyl carbon is observed [66]. In addition, Ivanov and coworkers have performed $^{13}$C SABRE experiments over a wide range of PTF values [52]. In their experiments, significant $^{13}$C SABRE hyperpolarisation was observed in a relatively strong PTF field of 90 mT. Therefore, it is likely that the efficiency of the INEPT approach could be greatly increased by carrying out the SABRE-polarisation transfer in a much stronger PTF and using a one-bond *J* coupling constant for the transfer.

In order to simplify the spectra in Figure 3a and to improve SNR, we applied broad-band $^1$H decoupling during $^{13}$C signal acquisition. Due to the anti-phase character of the SABRE-enhanced $^{13}$C NMR spectra, a refocusing delay was required prior to acquisition, as illustrated in the sequences in Figure 2c,d. The resultant SABRE-enhanced $^{13}$C{$^1$H} benchtop NMR spectra acquired with the refocused PA and INEPT pulse sequences are presented in Figure 3b. In both cases, the decoupling simplified the spectra and improved the signal-to-noise ratio (SNR), as expected. The narrow single resonances for each of the four $^{13}$C environments were well-resolved, including the very small difference of 0.5 ppm between the *para* (gray star) and *ortho* (blue square) carbon positions. The average SNR values for each $^{13}$C resonance of 4-MP, calculated for a set of repeat measurements acquired with each of the two detection methods, are given in Table 1. In contrast to the fully-coupled case, here the $^{13}$C{$^1$H} signal was much greater for the INEPT transfer when compared to the PA case. This implies that the delay used to refocus the anti-phase signals is insufficient to simultaneously refocus all of the signals in the PA case, leading to significant signal cancellation during $^1$H decoupling. Therefore, despite the apparent SNR advantage of PA detection in the fully-coupled case, the INEPT approach produces superior results for $^{13}$C{$^1$H} NMR spectra.

**Table 1.** Signal-to-noise ratios (SNR) for SABRE-enhanced $^{13}$C{$^1$H} NMR spectra of 4-MP acquired with the refocused PA and refocused INEPT pulse sequences. Values are the average over 7 (PA) and 3 (INEPT) repeat experiments.

| $^{13}$C Resonance | Refocused PA | Refocused INEPT |
|:---:|:---:|:---:|
| *para* | $9.9 \pm 0.5$ | $31 \pm 2$ |
| *ortho* | $18.9 \pm 0.9$ | $23 \pm 2$ |
| *meta* | $43 \pm 2$ | $91 \pm 7$ |
| *methyl* | $6.4 \pm 0.6$ | $73 \pm 7$ |

We note that all of these experiments were carried out in a PTF of 6.1 mT. The distribution of polarisation and the relative efficiency of the detection schemes will vary with the choice of PTF. Indeed, in previous work, the maximum PA $^{13}$C signal intensity was observed for a PTF equal to the Earth's magnetic field (~50 μT) [49]. In addition, as discussed above, it is probable that carrying out SABRE in a much higher PTF could be beneficial for optimising the hyperpolarisation of $^1$H directly bonded to $^{13}$C. This could significantly improve the overall efficiency of the INEPT approach.

### 3.2. SABRE-Enhanced 2D $^{13}$C–$^1$H Benchtop NMR Spectroscopy

In addition to the single-shot 1D $^{13}$C NMR experiments presented above, it is also of interest to consider the feasibility of acquiring SABRE-enhanced $^{13}$C–$^1$H 2D NMR spectra of samples at natural isotopic abundance. SABRE-enhanced 2D $^1$H–$^1$H benchtop NMR experiments have been reported previously [49]. Here we extend this approach to heteronuclear experiments, exemplified by a $^{13}$C-$^1$H HETCOR spectrum acquired using the pulse sequence in Figure 2f. Figure 5 presents a comparison between a 2D HETCOR benchtop NMR spectrum acquired using polarisation at thermal equilibrium for neat 4-methylpyridine (10.3 M, 16 scans, 307 min total experiment time, Figure 5a) and one acquired using SABRE hyperpolarisation of 260 mM 4-methylpyridine (1 scan, 69 min total experiment time, Figure 5c). The SABRE spectrum was achieved by re-hyperpolarising the solution outside of the spectrometer between the acquisition of each transient, as described previously [49].

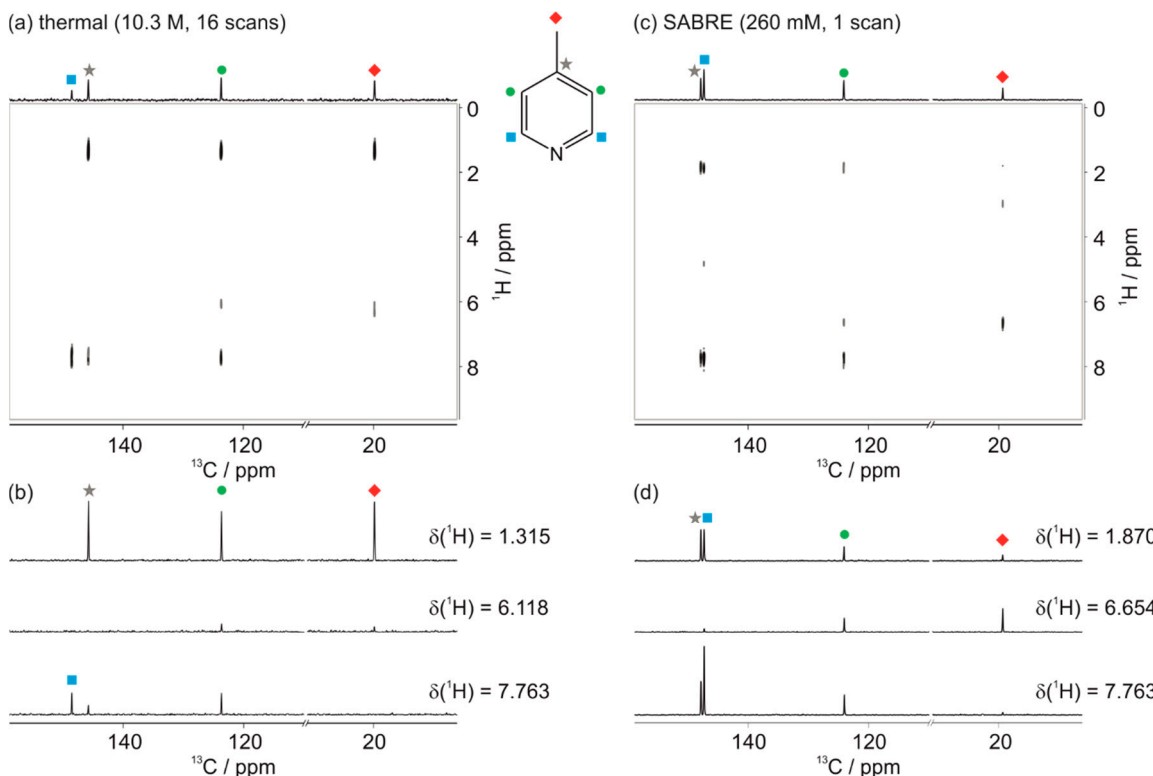

**Figure 5.** 2D $^{13}$C–$^1$H HETCOR benchtop NMR spectra of 4-methylpyridine (4-MP). (**a**) Thermally polarised spectrum of neat 4-MP (10.3 M) acquired with 64 steps each with 16 scans in a total experiment time of 307 min (5.1 h). (**b**) 1D slices through the 2D spectrum in (**a**) at the chemical shift of the *methyl* proton resonance (top), *meta* proton resonance (middle), and *ortho* proton resonance (bottom). (**c**) SABRE-hyperpolarised 2D spectrum of 260 mM 4-MP with 5.2 mM active SABRE catalyst in methanol-$d_4$ acquired with 90 steps each with a single scan in a total experiment time of 69 min. (**d**) 1D slices through the 2D spectrum in (**c**) as in (**b**). Note, the differences in chemical shifts between the two spectra are due to the presence of the solvent (methanol-$d_4$) in the SABRE case.

The 2D spectrum in Figure 5c demonstrates the ability of SABRE hyperpolarisation to enable high-sensitivity 2D $^{13}$C–$^1$H benchtop NMR on relatively low concentration samples at natural isotopic abundance and in reasonable experiment times. In order to obtain a comparable SNR, a neat sample (10.3 M) of 4-MP and 16 scans were used for the reference spectrum in Figure 5a. The use of hyperpolarisation allowed for a reduction in concentration by two orders of magnitude as well as a reduction in experiment time by a factor of ~4.5. We note that in order to achieve a comparable result to the SABRE spectrum at the lower concentration, signal averaging for over 300 days would have been required. As in the 1D INEPT case, efficient hyperpolarisation transfer from $^1$H to $^{13}$C was observed using a relatively long-range coupling constant of $J_{CH} = 10$ Hz due to the inefficiency of SABRE hyperpolarisation of protons directly bonded to $^{13}$C.

## 4. Discussion and Conclusions

In this work we have demonstrated the acquisition of high-resolution 1D and 2D $^{13}$C{$^1$H} benchtop NMR spectra of relatively low concentrations of the target analyte at natural abundance in a single scan. The SABRE-enhanced $^{13}$C{$^1$H} spectra are easily interpreted at 1 T, including the separation of peaks with chemical shift differences of less than 0.5 ppm. Using SABRE hyperpolarisation with a PTF of 6.1 mT, the optimal SNR was achieved by the PA approach in the fully-coupled spectra but the refocused INEPT approach provided optimal SNR in the decoupled spectra. The superior performance of the INEPT sequence is likely due to the wide range of anti-phase $^{13}$C–$^1$H terms that are excited in the PA approach and that are not easily refocused using a single delay. While the INEPT approach

provided the highest SNR for the $^{13}$C{$^1$H} spectra, our results highlight that $^1$H nuclei directly bonded to $^{13}$C are not hyperpolarised efficiently under these experimental conditions. In future work, adopting the approach of Ivanov and coworkers [52] to perform SABRE in a much higher PTF within the bore of the benchtop NMR spectrometer could lead to much more efficient $^1$H–$^{13}$C hyperpolarisation via a one-bond INEPT transfer.

One of the key advantages of the SABRE approach over other hyperpolarisation methods for analytical applications is that it is reversible and so a single sample can be re-polarised multiple times. We have exploited this feature to acquire 2D $^{13}$C–$^1$H benchtop NMR spectra, with re-polarisation achieved outside of the spectrometer between each step of the 2D experiment. However, in our flow-based approach to SABRE-enhanced 2D NMR, evaporation of the solvent during sample transfer and $p$-H$_2$ bubbling ultimately limits the maximum number of transients that can be achieved for a single sample. In addition, the transfer of the sample between the spectrometer and the mixing chamber is time-consuming relative to the other steps in the experiment. These limitations could potentially be overcome by using more efficient sampling methods, such as the single-shot 2D methods [67], which have previously been demonstrated using SABRE and high-field detection [68]. Alternatively, hyperpolarisation within the bore of the benchtop spectrometer, as suggested above, would significantly limit the transfer time and distance between the SABRE-polarisation transfer step and the signal detection step. This approach has the potential to make SABRE-enhanced 2D $^{13}$C benchtop NMR viable for more routine applications by significantly decreasing experiment times, increasing the maximum number of transients, and improving sensitivity by increasing the efficiency of the INEPT transfer.

SABRE hyperpolarisation was achieved here using a model analyte, 4-methylpyridine. In order for an analyte to be strongly enhanced by SABRE, it needs to reversibly bind to the catalyst on an appropriate timescale. The residence time on the catalyst must be long enough for significant polarisation transfer to occur but not too long such that NMR relaxation dominates. It is well established that N-heterocycles are good SABRE substrates. However, recent advances in SABRE catalysis have extended this approach to other functional groups, such as amines, using the standard SABRE mechanism [69]. In addition, a new mechanism for transfer, called SABRE-Relay, has been introduced, whereby a carrier with exchangeable protons is hyperpolarised through direct association to the catalyst and then transfers polarisation to a target substrate via proton exchange. In principle, this provides a route to the hyperpolarisation method of any target substrate with exchangeable protons [69]. Thus, the range of target analytes that are accessible to the SABRE approach are expected to increase going forward.

**Supplementary Materials:** The following are available online at http://www.mdpi.com/2076-3417/9/6/1173/s1, Figure S1: Schematic and photo of the automated flow system used for SABRE hyperpolarisation with benchtop NMR detection, Table S1: List of the variable RF pulse angles used in the single-shot hyperpolarisation lifetime measurements.

**Author Contributions:** Conceptualization, M.E.H.; methodology, A.D.R., P.M.R., and M.E.H.; formal analysis and investigation, A.D.R., P.M.R., and M.E.H.; writing—original draft preparation, A.D.R. and M.E.H.; writing—review and editing, A.D.R., M.E.H., and P.M.R.; supervision and funding acquisition, M.E.H.

**Funding:** This research was funded by the UK Engineering and Physical Sciences Research Council (EPSRC), grant numbers EP/M020983/1 and EP/R028745/1.

**Acknowledgments:** The authors thank Richard John and Victoria Annis for technical support and Simon Duckett for useful discussions.

**Conflicts of Interest:** The authors declare no conflict of interest. The funders had no role in the design of the study; in the collection, analyses, or interpretation of data; in the writing of the manuscript, or in the decision to publish the results.

**Data Access Statement:** All experimental NMR data reported in this work is available via Research Data York at http://dx.doi.org/10.15124/419d378a-764c-4a25-a064-8a1a7622d4d8.

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
