# Peer review of "Hyperpolarised 1H–13C Benchtop NMR Spectroscopy"

_applsci, doi:10.3390/app9061173_

Reviewer 1 Report

The authors showed hyperpolarized 1H-13C NMR experiments (refocused INEPT and HETCOR) of 4-methylpyridine using low-magnetic field benchtop NMR instrument. Hyperpolarized NMR signals are effective for detection of organic- and bio-molecules. In general, elegant detections of hyper polarized NMR signals are quite difficult. In this study, there are beautiful polarized NMR signals by 2D experiments even in benchtop instrument. The manuscript are well written. Therefore, this manuscript is suitable for the Applied Science.

minor comments

1. The authors should increase citation list;

Hyperpolarized 13C allows a direct measure of flux through a single enzyme-catalyzed step by nor

Matthew E. MerrittCrystal HarrisonCharles StoreyF. Mark JeffreyA. Dean Sherry, and Craig R. Malloy

Rapid sample injection for hyper polarized NMR spectroscopy  

Sean Bowen and Christian Hilty 

Phys. Chem. Chem. Phys. 2010, 12, 5766-5770

Author Response

The suggested references have been added to the text.

Reviewer 2 Report

The science in this paper is very strong and I strongly recommend publishing it after addressing these points:

1- In the introduction, the authors mention some NMR basics about losing sensitivity and resolution upon a decrease in the strength of the magnetic field (and polarization accordingly), I believe it's important to mention this in terms of numbers and equations (like the resolution goes linear with the magnetic field while the sensitivity is proportional to B˚^3/2)

2- The polarization transfer from p-H2 needs also to be illustrated in terms of percentage or numbers (for example using DNP, the increase in sensitivity is coming from the fact that electrons have more than 600 times the gyromagnetic ratio of protons). How is that for p-H2?

3- In the Materials and Methods, what dictates the value of the PTF (it is mentioned that this ranges from 0-20 mT)? Needs clarification

4- What paramagnetic catalysis was used for the H2 gas? Lanthanides?

5- In figure 2b, looking at this pulse sequence, this doesn't look for me like typical INEPT where there is 180 refocusing in the middle on both atoms and you end up with the anti-phase term on the heteroatom? Can you clarify or discuss this in a phrase or two?

6- In the caption of figure 3, I see "all spectra were acquired in a single scan", is this correct?? You apply only ONE scan to get these spectra? To be sure we are talking about the same thing: this is the number of times the pulse sequence is executed with interscan delay (d1)?

7- Eventually, I would like to see in terms of numbers comparison of s/n from this technique vs measuring on a 400 or 500 MHz magnet? Why would I go to a more complicated technique like that to measure 1D 1H or 13C NMR spectra if there's easily 400 or 500 MHz magnets in most NMR-labs now?

Author Response

1 - These clarifications have been added to the introduction.

2 -  In these experiments the source of the polarisation is pH2 and the level of pH2 enrichment (>99%) is given in the methods section. In the manuscript, we have not reported quantitative enhancement factors or polarisation levels. This is a consequence of the fact that thermally polarised 13C reference measurements cannot be obtained on the given sample concentrations at natural isotopic abundance even after days of signal averaging. Therefore it was not feasible to calculate meaningful and quantitative enhancement levels. We have supplied SNR to provide readers with quantification of the signal obtained.

3 - As described in the introduction to the paper, the optimum value of the PTF is dictated by many factors, particularly the chemical shfits and J coupling values of the active SABRE complex. While we have not studied the PTF dependence in this work it has been well explored in the literature. We have added a relevant reference to the materials and methods section to clarify this point.

4 - The home-built parahydrogen generator contains a paramagnetic catalyst based activated charcoal as described previously in Ref. 65. This reference has been added to the materials and methods section.

5 - In the refocusing pulse sequences, it was found for both hyperpolarised and thermally polarised experiments that the performance was better in the absence of the 180 deg refocusing pulses. This unexpected result was attributed to B1 field inhomogeneity. A statement has been added to clarify this in the text. 

6 - Each spectrum in Figure 3 was acquired in a separate experiment, each of which was completed using a single scan. To clarify this in the text we have changed the word "all" to "each" in the Figure 3 caption.

7 - The focus of this paper was on the feasibility of 13C detection using benchtop NMR with SABRE hyperpolarisation. SABRE-enhanced 13C NMR has been explored at high field previously. The benefits of benchtop NMR are primarily cost and portability. While it has been established (see ref  65 for example) that polarisation levels in SABRE are independent of detection field, due to more favourable dectection at high field, we would expect SNR to be higher using this same technique with a 400 or 500 MHz spectrometer for detection. However these direct comparisons were not made in this work.

Reviewer 3 Report

The authors present a practical paper on implementing the hyperpolarization method of SABRE for 13C NMR with a benchtop NMR spectrometer. There is clearly more work ahead in this area before SABRE can be more universally applied in benchtop NMR spectrometers. The manuscript is contributing useful insight for such endeavor and deserves therefore publication. The manuscript is well written and I found very few aspects to consider for final revision. Essentially, to the best of my judgement, the manuscript could be published as is.

Comments:

The authors claim on page 7 that polarization of protons directly bonded 13C is found less efficient because the PTF was too low. Why did the authors not test this theory by applying a stronger than the 6.5 mT field strength? The authors refer on lines 324-326 for future work, but it is unclear why the PTF could not be tweaked to be at list a little bit stronger, just to see if this would increase the polarization transfer efficiency.

On page 9, second paragraph, the authors state that more explorations of the SABRE process might be possible with the benchtop NMR, without being specific. It is unclear to me what the authors have in mind and I find this paragraph vague.   

A couple of typographical corrections:

Line 114 such as (not such at)

Line 339, potential to make

Author Response

1 - The PTF required to meet the one-bond 1H-13C transfer condition ~100 mT is more than an order of magnitude larger than the PTF of 6.1 mT used in this work. Therefore it is not a small change to the experimental protocol to carry-out these experiments. (For reference, the maximum output of the electromagnet used for the automated experiments is only 14 mT). It is for this reason that these experiments were designated for future work.

2 - In the discussion of Figure 5 we have mentioned that the 2D correlation spectroscopy approach could potentially be used to elucidate the different spin states hyperpolarised by the SABRE process through a comparison of the cross peaks present in the hyperpolarised spectrum compared to the thermal reference. However we recognise that this point is not well described in the text and is not a major conclusion of this work. Therefore it has been removed to improve the overall clarity of the paper.

3 - The highlighted typos have been corrected.

Reviewer 4 Report

Clearly written report explaining the use of (1) SABRE-driven hyperpolarization (2) on a model system (3) for 13C detection (4) using a benchtop NMR spectrometer. The results are well supported, hypotheses are explained and justified, and the conclusions are sound. Several minor comments:

lines 112-113: authors describe some limitations of their automated SABRE flow system. Are these engineering limitations or fundamental limitations? Possible point to highlight for the reader.

line 114: "...control over parameters such *as* the duration of..." [original reads "at"]

line 115-116: Can the authors comment on the uncertainty introduced into 2D spectra when each scan in the indirect dimension was measured using a potentially different quantity of hyperpolarization? They reference reproduciliblilty, but there is no quantitative measure. Even if it is in Ref 47, a quick numerical value should be mentioned here. This may also affect results shown in Fig. 5c (though, looking at the spectrum, it seems the shot noise is small)

lines 130-133: Confusion about the procedure here. Six shake-and-drop experiments completed within 10 minutes, each with renewed pH2, followed by a five-minute equilibration period. Why were the shake-and-drop experiments repeated? Was the magnetization monitored through the six shake-and-drops? Was 5 minutes necessary for bubble to dissipate? Was significant  magnetization lost during the 5-min equilibration period? Was this entire procedure done only once before *all* quantitative experiments were made, or was it repeated before each quantitative measurement? If the 5-minute equilibration period was to allow the solution to settle/bubble to dissipate, how long must the experimenter wait after adding fresh pH2 to the active-catalyst-containing solution before collecting data? Similar questions arise in lines 149-151 for the automated pH2 delivery system. This passage would benefit from some clarity/explanation as to the experimenters' rationale.

lines 219-251: A really nice explanation to resolve the apparent contradiction in hyperpolarization lifetimes brought up in lines 196-213.

line 286: Should this in-text figure ref be to Figure 5c? (currently 5b)

line 346: For the benchtop instrument used in these experiments, what is the approximate timescale for the residence on the catalyst? A number, if available, would be helpful to readers.

Refs. 47, 60, and 62 are incomplete

Author Response

1. All of the mentioned typos have been corrected.

2. Yes the limitations of the automated system are largely engineering limitations. This has been clarified in the text.

3. The reproducibility in the automated system is expected to be no more than 5% (as in the quoted reference). This has been added to the text.

4. The description of the activation procedure has been clarified in the text.

5. With regards to the residence time of the substrate on the catalyst, this is sample rather than instrument dependent. This parameter was not explicitly measured for the substrate and catalyst system in this study.